# Hydrogen-Rich Water Mitigates LPS-Induced Chronic Intestinal Inflammatory Response in Rats via Nrf-2 and NF-κB Signaling Pathways

**DOI:** 10.3390/vetsci9110621

**Published:** 2022-11-08

**Authors:** Jin Peng, Qi He, Shuaichen Li, Tao Liu, Jiantao Zhang

**Affiliations:** 1Heilongjiang Key Laboratory for Experimental Animals and Comparative Medicine, College of Veterinary Medicine, Northeast Agricultural University, Harbin 150038, China; 2Institute for Genome Biology, Research Institute for Farm Animal Biology (FBN), Wilhelm-Stahl-Allee 2, 18196 Dummerstorf, Germany

**Keywords:** hydrogen-rich water, LPS, oxidative stress, gut inflammation, NF-κB pathway, Nrf-2 pathway

## Abstract

**Simple Summary:**

Our study provides substantial evidence that hydrogen-rich water can prevent intestinal damage caused by LPS-induced stimulation. Intestinal injury caused by LPS is related to the NFκB signaling pathway. LPS promotes the generation of inflammatory factors and oxygen free radicals by activating the NFκB signaling pathway in intestinal cells, causing inflammation and oxidative stress and destroying the tight-junction state of the intestine, causing intestinal damage. Hydrogen-rich water can activate the Nrf-2 signaling pathway to remove excess oxygen free radicals and relieve oxidative stress. The Nrf-2 signaling pathway, an upstream signaling pathway of NFκB, can act as a protective mechanism against inflammation by regulating NFκB. In summary, hydrogen-rich water can inhibit oxidative stress and regulate NF by activating Nrf-2 signal pathway κB signal pathway, inhibit the production of inflammatory cytokines, maintain the tight connection in the intestine, and protect the intestine from damage caused by lipopolysaccharide. Our research contributes to further understanding of the mechanism of hydrogen-rich water, as well as its value for clinical application.

**Abstract:**

Long-term exposure to low-dose lipopolysaccharide can impair intestinal barriers, causing intestinal inflammation and leading to systemic inflammation. Hydrogen-rich water possesses antioxidant and anti-inflammatory functions and exerts inhibitory effects on various inflammatory diseases. In this study, we investigated whether oral hydrogen-rich water could prevent lipopolysaccharide-induced chronic intestinal inflammation. An experimental model was established by feeding hydrogen-rich water, followed by the injection of lipopolysaccharide (200 μg/kg) in the tail vein of rats after seven months. ELISA, Western blot, immunohistochemistry, and other methods were used to detect related cytokines, proteins related to the NF-κB and Nrf-2 signaling pathways, and tight-junction proteins to study the anti-inflammatory and antioxidant effects of hydrogen-rich water. The obtained results show that hydrogen-rich water significantly increased the levels of superoxide dismutase and structural proteins; activated the Nrf-2 signaling pathway; downregulated the expression of inflammatory factors cyclooxygenase-2, myeloperoxidase, and ROS; and decreased the activation of the NF-κB signaling pathway. These results suggest that hydrogen-rich water could protect against chronic intestinal inflammation in rats caused by lipopolysaccharide-induced activation of the NF-κB signaling pathway by regulating the Nrf-2 signaling pathway.

## 1. Introduction

The Nrf-2 and NFκB signaling pathways are known to cause various diseases. Inflammatory responses are tightly regulated by pro- and anti-inflammatory mediators, critical factors affecting the development of inflammatory bowel disease (IBD), obesity, and neurological and systemic diseases [1]. Chronic inflammation is a predisposing factors for chronic conditions, such as metabolic disorders and cancer. Studies have shown that antioxidant genes activate under stress and protect cells via the Nrf-2 signaling pathway [2]. Therefore, anti-inflammatory and/or antioxidant molecular therapy might be a suitable approach to reduce or prevent the progression of inflammatory diseases.

A significant part of food digestion occurs in the small intestine, which makes up a large portion of the body’s internal surface. The functional and morphological integrity of the small intestine is vital to prevent toxic substances from damaging the tight-junction barrier. The dysregulation of the gut barrier can result in several pathological conditions in the intestines, such as the accumulation of endotoxin, which could result in an inflammatory response in the intestine [3]. Damage to the intestinal barrier disrupts the metabolic balance of the intestine and dysregulates the intestinal microbiota, leading to an enhanced inflammatory response [4,5].

Tight-junction proteins are connected via multiprotein complexes. These tight-junction proteins are located in the apical portion of the lateral membrane of epithelial cells. During physiological activities, tight-junction proteins protect the solute from the surroundings via the intercellular space [6]. Tight junctions regulate the process of ion, water, and solute transport by the paracellular pathway and blockage of immunogenic macromolecules; only controlled and selective movement is allowed during passive osmosis [7]. Tight-junction protein disorder leads to the destruction of the mucous layer and increases intestinal permeability. This phenomenon leads to an influx of bacterial endotoxins into the bloodstream from the damaged epithelial barrier and produces various inflammatory responses. This process results in persistent chronic inflammation in the gut and the dysregulation of the gut as a physiological protector against exogenous substances. Endotoxin-induced systemic inflammation produces macrophage infiltration into other tissues and promotes the release of proinflammatory mediators [8]. The inflammatory cascade leads to the dysfunction of various tissues and permanent lesions [9]. Therefore, the structural integrity of the intestinal barrier is crucial with respect to maintaining its barrier function and immune-metabolic homeostasis.

Lipopolysaccharides (LPS) (endotoxin) are the main component of the outer membrane of Gram-negative bacteria. They induce cellular apoptosis, increase epithelial cell permeability, and produce DNA damage. Prolonged exposure to LPS has been shown to activate the proinflammatory response mediated by the NF-κB pathway, thereby compromising the intestinal barrier [10]. Therefore, LPS is widely used to study animal models of inflammation [11,12]. LPS-induced inflammation disrupts the structure and function of the intestinal epithelium. Recent studies have shown that the LPS-induced intestinal permeability model regulates the expression of tight-junction proteins in the epithelial cells of the small intestine [13].

There has been a growing interest in hydrogen (H2), owing to its numerous biological activities, such as antioxidant properties, as per the results of animal testing [14]. Ohsawa et al. reported [15] that H2 could protect cells from oxidative stress damage by scavenging cytotoxic oxygen free radicals. H2 molecules exhibit significant anti-inflammatory properties [16]. Buchholz et al. found that hydrogen therapy inhibited the inflammatory response in a rat model of small intestine transplantation [17]. Chen et al. confirmed that hydrogen therapy exerted a protective effect on acute pancreatitis by inhibiting the activation of NF-κB and oxidative stress [18]. When H2 was administered under inflammatory conditions, it affected various intracellular pathways, including NF-κB, ERK, P38, JNK, and Nrf-2. It also showed effects at the genetic level, such as the expression of inflammatory cytokines [19]. Japanese researchers were the first to discover hydrogen-rich water (HRW) as a new antioxidant, which can be generated by dissolving hydrogen in water [15]. HRW showed expected protective effects on intestinal injury caused by inflammation, sepsis, septic shock, and radiation by upregulating the expression of tight-junction proteins, reducing the release of inflammatory factors, activating the Nrf-2 signaling pathway, and inhibiting oxidative stress [20,21,22].

HRW has exhibited various biological roles as a therapeutic antioxidant, such as anti-inflammatory and antioxidant effects. Therefore, the aim of this study was to explore whether oral administration of HRW could alleviate inflammatory responses in LPS-induced rat intestinal inflammation.

## 2. Materials and Methods

### 2.1. Animals

Sprague–Dawley (SD, n = 48) rats weighing 220 ± 20 g (male = 24; female = 24) were procured from Baikui Tianxing Farm, Hulan District, Harbin City. The rats were raised in the Animal Breeding Center of Northeast Agricultural University under a controlled temperature of 20 ± 1 °C and at a relative humidity of 55–60% with a 12 h/12 h light/dark cycle. Animals had free access to water and were fed a standard diet. All rats were allowed to acclimatize for one week before starting the experiments. This study was approved by the Ethical Committee for Animal Experiments NEAUC2020 04 17 (Northeast Agricultural University, Harbin, China).

### 2.2. Drug Concentration

Hydrogen generated by a hydrogen generator (QL-500, Shandong Saikesaisi Hydrogen Energy Co., Ltd., Jinan, China) was stored in a bottle and soaked in 400 mL of distilled water at a rate of 500 mL/min using the exhaust method. It took 15 min to reach saturation, resulting in hydrogen-rich [22].

LPS injection (Shanghai Xinle Biotechnology Co., Ltd., Shanghai, China) was diluted using sterile deionized water to a final concentration of 200 μg/kg before use [23].

### 2.3. Experimental Design

Rats were randomly assigned to four experimental groups (n = 12, half male and half female). Drinking time was limited at specific time points (8:00 to 10:00, 13:00 to 15:00, and 18:00 to 20:00). All rats had free access to food. A: The control group (CON) was fed with distilled water for seven months; after seven months, sterile normal saline (concentration: 0.5 mL/kg body weight) was injected via the tail vein on the first day of every week. B: The hydrogen-rich water group (HRW) was fed HRW for seven months; after seven months, sterile saline was injected into the tail vein on the first day of the week in the same concentration as that of the CON group. C: The tail vein injection of lipopolysaccharide (LPS) group was fed with distilled water for seven months; after seven months, LPS (concentration: 200 μg/kg body weight) was injected into the tail vein on the first day of every week. D: The HRW plus lipopolysaccharide injection through the tail vein (HRW + LPS) group was HRW for seven months; after one month, LPS was injected into the tail vein on the first day of every week in the same concentration as that of the LPS group. All rats underwent a tail vein blood draw every weekend for four weeks.

### 2.4. Sample Collection

Weight changes in the rats were recorded weekly. After four weeks, one hour before sacrifice, the rats were administered ovalbumin (OVA) by gavage, the concentration of OVA was 250 mg·kg-1·bw-1 one hour after gavage. The permeability of the intestine to macromolecular substances was assessed by measuring OVA in serum [24]. The rats were euthanized under general anesthesia. Blood was collected from the heart and centrifuged (10 min, 12,000× *g*, 4), and serum was stored at −80 °C. The ileum was removed, and the intestinal contents were washed with PBS and divided into two parts; one part was fixed with 10% formalin for histological, immunofluorescence, and immunohistochemical analysis, and the other part was used for ELISA, RNA extraction, real-time quantification, and Western blotting.

### 2.5. RNA Extraction and Quantitative Real-Time PCR (qRT-PCR)

Total RNA was isolated from ileum tissues using Trizol reagent (Invitrogen, Waltham, MA, USA). The RNA concentration and purity of all samples were determined on a NanoDrop spectrophotometer (Thermo, Wilmington, DE, USA). Reverse RNA transcription was performed using a reverse transcription kit (Nearshore Protein Technology Co., Ltd. Suzhou, China). Sangon Bioengineering (Shanghai, China) Co., Ltd. designed the primers. Homology searches were performed using BLAST from the NCBI (https://www.ncbi.nlm.nih.gov). After verification, the primers were synthesized by Sangon Bioengineering Company at a concentration of 10 μM. qRT-PCR was performed using a Roche Light Cycler 480 real-time PCR instrument (Roche, Basel, Switzerland), and each sample was run in 3 replicates. Data were collected, and the relative mRNA expression of target genes was assessed using the 2^−ΔΔ Ct^ method. Table 1 provides a list of primers and their sequences.

### 2.6. ELISA

The expressions of IL-1β, IL-6, TNF-α, MCP-1, cyclooxygenase-2 (COX-2), myeloperoxidase (MPO), ovalbumin (OVA), malondialdehyde (MDA), and superoxide dismutase (SOD) in the ileum tissue were determined using an ELISA kit (Nanjing Jiancheng Bioengineering Institute, Nanjing, China). The absorbance (OD value) was measured at a wavelength of 450 nm using an Epoch microplate reader (BioTek, Winooski, VT, USA), and the sample concentration was calculated.

### 2.7. Western Blot Analysis

The total protein was extracted from ileum tissue samples using RIPA lysis buffer (Biyuntian Biotechnology Co., Ltd., Shanghai, China). The protein concentration was determined using a BCA protein assay kit (Biyuntian Biotechnology Co., Ltd., Shanghai, China). SDS-PAGE gels were prepared on glass plates using an SDS-PAGE gel kit (Biyuntian Biotechnology Co., Ltd., Shanghai, China). A volume of 5 μL of protein samples was loaded per well for SDS-PAGE and transferred to polyvinylidene fluoride membranes. Membranes were blocked with 5% nonfat dry milk (Wandashan Dairy Co., Ltd., Heilongjiang, China) in 1 × TBST for 120 min and incubated in corresponding primary antibodies (TLR4, MyD88, IκBα, p-IκBα, p50, p-p50, p65, p-p65, Nrf-2, HO-1, and NQO1), then incubated overnight at 4 °C. We analyzed the effects of HRW and LPS on the NF-κB and Nrf-2 signaling pathways by detecting changes in these proteins. The next day, samples were washed three times with one × TBST and incubated with horseradish peroxidase (HRP)-labeled secondary antibody (dilution ratio 1:10,000) on the membrane for two h on a shaker at room temperature. Then, the samples were washed thrice with one × TBST again, followed by color development using ECL chemiluminescence (Biyuntian Biotechnology Co., Ltd., Shanghai, China). Images were collected using a Tanon-5200 chemiluminescent imaging system (Tianneng Technology Co., Ltd., Shanghai, China). The protein bands were analyzed and quantified using ImageJ software (NIH, Bethesda, MD, USA).

### 2.8. Histopathology

The ileum pathologically analyzed to observe the intestinal structure, the arrangement of the ileal villi, the presence of lesions, and the infiltration of inflammatory cells to evaluate the protective effect of hydrogen-rich water on the intestinal injury. After the rats were sacrificed, intestinal tissues were preserved in a 10% formalin fixative solution, removed, placed in cassettes for slicing, and rinsed overnight in running water. The samples were dehydrated through gradient alcohol and cleared using xylene. Next, the samples were immersed in wax, embedded in paraffin, and cut into five μm thick sections. These sections were stained with hematoxylin and eosin for histopathological analysis. The samples were sliced in a crosscut ring direction.

### 2.9. Immunohistochemistry 

The levels of ZO-1, occludin, and claudin-1 were detected by immunohistochemistry, and the infiltration of macrophages was determined by detecting CD68. Immunohistochemistry was performed on five μm thick sections after deparaffinization following the established protocol. The sections were dehydrated in conventional gradient alcohol, cleared with xylene, and washed with PBS. The sections were incubated in proteinase K (20 μg/mL) for 20 min at room temperature, followed by rinsing with TBS. Endogenous peroxidase activity was blocked by incubating the slides with 3% H_2_O_2_ for 15 min. The slides were blocked with 3% BSA. Then, samples were diluted 1:100 with primary anti-IgG antibody (ZO-1, occludin, claudin-1, and CD68) overnight. Subsequently, sections were washed with wash buffer, followed by coincubation with 3,3-O-diaminobenzidine (DAB) solution and horseradish peroxidase (HRP) in the dark. Each sample was left at room temperature for 30 min. Sections were counterstained with hematoxylin, dehydrated using xylene, and mounted in DPX. Slides were observed under a light microscope. The images were acquired at 150× magnification; yellow or brown–yellow indicated positive staining, with three dyes per group, and stain intensity was quantitated using ImageJ software (NIH, Bethesda, MD, USA).

### 2.10. Immunofluorescence

ROS content was detected in rat ileum by immunofluorescence. The slides were dewaxed in xylene and rehydrated using graded ethanol. Next, the sections were incubated with anti-ROS antibody (1:1000) (Abcam Inc., Cambridge, MA, USA) for 1 h at room temperature, followed by 30 min incubation with a secondary antibody and a brief wash. Slices were observed using a fluorescence microscope.

### 2.11. Statistical Analysis

Results were analyzed and graphed using Prism 7.0 software (GraphPad Software, San Diego, CA, USA). The results were expressed as mean ± standard deviation (mean ± SD). A Student’s *t*-test or one-way ANOVA test was performed to determine significant differences; *, # *p* < 0.05; **, ## *p* < 0.01.

## 3. Results

### 3.1. Effect of HRW on Gut Permeability

Rat gut structure was histologically analyzed to investigate the integrity of the gut barrier. The ileum structure of the CON group was complete and transparent (Figure 1A). The villi were also neatly arranged, and no pathological changes were observed, such as inflammation or edema. In the HRW group, the intestinal villi were tidier and occasionally shed, and the structure was relatively intact. In the LPS group, the structure of the intestinal villi was loose, disordered, and even shed. There was also a large number of infiltrated lymphocytic cells. Compared with the LPS group, the structure of intestinal villi was significantly improved in the HRW + LPS group, and the villi of the small intestine were slightly damaged and closely arranged.

The integrity of the intestinal barrier structure forms the basis of intestinal function. Here, the effect of HRW on gut integrity was examined by measuring OVA concentrations and tight-junction protein expression. The expression of tight-junction proteins is a crucial indicator of intestinal mucosal integrity. The expression of tight-junction proteins in ileal tissue was determined. Compared with the CON group, the expressions of ZO-1, occludin, and claudin-1 were significantly reduced in the intestinal tissue of the LPS group (Figure 1B, *p* < 0.01). Compared with the LPS group, the expressions of ZO-1, occludin, and claudin-1 increased in the HRW + LPS group (Figure 2, *p* < 0.01). Additionally, the OVA content was significantly increased in the LPS group compared with the CON group (Figure 1C, *p* < 0.01). In contrast, the OVA content was significantly decreased in the HRW + LPS group (Figure 1C, *p* < 0.01).

### 3.2. Effects of HRW on Inflammatory Cytokines in the Gut of Rats with Chronic Inflammation

Compared with the CON group, IL-1β, IL-6, TNF-α, and MCP-1 levels in serum and tissues were significantly elevated in the LPS group (Figure 2A,B, *p* < 0.01). However, the concentrations of IL-6 and TNF-α in the HRW + LPS group were decreased in the serum compared with the LPS group (Figure 2A, *p* < 0.05). Additionally, the concentrations of IL-1β and MCP-1 were significantly reduced (Figure 2A, *p* < 0.01). In the intestinal tissue, compared with the LPS group, the HRW + LPS group had decreased the levels of IL-1β and TNF-α (Figure 2B, *p* < 0.05), and the expression of IL-6 and MCP-1 was decreased significantly (Figure 2B, *p* < 0.01). qPCR data showed that the relative gene expression of inflammatory cytokines in rat ileal tissue was significantly upregulated after LPS treatment (Figure 2C, *p* < 0.01). The expression of cytokine-related genes in the tract tissue was significantly inhibited, and the relative gene expression of cytokines in the HRW + LPS group was significantly downregulated compared with the LPS group (Figure 2C, *p* < 0.01). In this study, we determined the content of COX-2 in rat ileal tissue, which responded to inflammatory responses in rat intestinal tissue. Compared with the CON group, the COX-2 content in the tissues of the LPS group was significantly enhanced (Figure 2D, *p* < 0.01). However, the COX-2 content in the tissues of the HRW + LPS group was significantly decreased compared with the LPS group (Figure 2D, *p* < 0.01).

### 3.3. Effects of HRW on Macrophage Infiltration in the Gut of Rats with Chronic Inflammation

Compared with the CON group, the number of macrophages in the intestinal tissue was significantly increased in the LPS group (Figure 3A, *p* < 0.01). In contrast, the HRW + LPS group had a significantly decreased number of macrophages in ileal tissue compared with the LPS group (Figure 3A, *p* < 0.01).

### 3.4. Effects of HRW on the NF-κB Signaling Pathway in Rats with Chronic Inflammation

In this study, the expression of NF-κB signaling pathway-related proteins was measured in each group (Figure 4A). Compared with the CON group, the protein expressions of TLR4 and MyD88 in the LPS group were significantly increased (Figure 4B,C, *p* < 0.01). In contrast, the protein expressions of TLR4 and MyD88 in the HRW + LPS group were lower compared with those in the LPS group (Figure 4B,C, *p* < 0.05). The expression of IκBα protein was significantly reduced in the LPS group (Figure 4D, *p* < 0.01), but it was increased in the HRW + LPS group (Figure 4D, *p* < 0.05). HRW pretreatment upregulated the expression of IκBα protein (Figure 4D, *p* < 0.05), and p-IκBα protein expression was significantly increased in the HRW + LPS group compared with LPS group (Figure 4D, *p* < 0.01). The expressions of p-p50 and p-p65 in the LPS group were significantly increased (Figure 4E,F, *p* < 0.01), whereas the expression of p-p50 and p-p65 in HRW + LPS group was inhibited (Figure 4E,F, *p* < 0.05).

### 3.5. Effects of HRW on Oxidative Stress-Related Indicators in Rats with Chronic Inflammation

ROS content was determined in intestinal tissue by immunofluorescence technique (Figure 5A). Compared with the CON group, the LPS group showed significantly increased ROS levels in the intestinal tissue (Figure 5A, *p* < 0.01). Compared with the LPS group, the HRW + LPS group had significantly reduced ROS levels (Figure 5A, *p* < 0.01). Furthermore, MPO levels were measured in rat intestinal tissue. Compared with the CON group, the MPO content in the LPS group was significantly increased (Figure 5B, *p* < 0.01), and compared with the LPS group, the MPO content was significantly decreased in the HRW + LPS group (Figure 5B, *p* < 0.01).

Compared with the CON group, the SOD content in the tissues in the LPS group was decreased (Figure 5C, *p* < 0.05), whereas the SOD content in the HRW group was significantly increased (Figure 5C, *p* < 0.01). Compared with the LPS group, the SOD content in the HRW + LPS group was significantly increased (Figure 5C, *p* < 0.01). Compared with the CON group, the MDA content in the LPS group was significantly enhanced (Figure 5D, *p* < 0.01). Compared with the LPS group, the MDA content in the HRW + LPS group was significantly decreased (Figure 5D, *p* < 0.01).

### 3.6. Effects of HRW on the Nrf-2 Signaling Pathway in Rats with Chronic Inflammation

The expression of Nrf-2 signaling pathway-related proteins was measured in each group (Figure 6). Compared with the CON group, HRW treatment significantly increased Nrf-2, HO-1, and NQO1 protein levels (Figure 6B–D, *p* < 0.01), and the expressions of HO-1 and NQO1 protein decreased in the LPS group (Figure 6C,D, *p* < 0.01, *p* < 0.05). Compared with the LPS group, Nrf-2, HO-1, and NQO1 protein levels were significantly increased in the HRW + LPS group (Figure 6B–D, *p* < 0.01, *p* < 0.01, *p* < 0.05). 

## 4. Discussion

In recent years, hydrogen molecules have been shown to exhibit antioxidative and anti-inflammatory effects in research on human and animals [25], allowing researchers to explore the potential clinical application of hydrogen molecules in various diseases. Hydrogen molecules can be administered by multiple routes, including inhalation, oral administration, and parenteral administration. In this study, HRW was orally administered. Studies have shown that LPS stimulation can significantly disrupt intestinal villus morphology and enhance the relative expression of proinflammatory cytokines [26]. The results of the present study suggest that HRW alleviated LPS-induced intestinal injury by inhibiting oxidative stress and inflammation, activating the Nrf-2 signaling pathway, inhibiting the NF-κB signaling pathway, and upregulating the expression of tight-junction proteins. 

Previous studies have shown that many inflammatory processes induced by LPS cause oxidative damage [27]. Imbalanced oxidative status is associated with inflammation. Therefore, inhibition of oxidative stress could facilitate the inhibition of inflammation. Nrf-2 binds to antioxidant response elements and forms a core pathway regulating the endogenous antioxidant system [28]. This pathway is regulated by Keap1, with most Nrf-2 retained in the cytoplasm by Keap1 and degraded through ubiquitination; only a small portion of Nrf-2 enters the nucleus to regulate the expression of antioxidant enzymes. When the antioxidant defense system is activated, Keap1 is decoupled from Nrf-2. Subsequently, cysteine modifications induce conformational changes in Keap1, whereas Nrf-2 is transported across the nucleus. Then, Nrf-2 forms heterodimers with the small-molecule musculoaponeurotic fibrosarcoma (Maf) proteins and interacts with upstream molecules, AREs, to regulate the transcription of downstream target genes. Nrf-2 binds to target genes GCLC, GCLM, HO-1, and NQO-1 through antioxidant response elements and regulates the expression of related antioxidant enzymes, thereby reducing ROS levels [29,30,31,32,33,34,35,36], reducing ROS-mediated intestinal damage, and maintaining redox balance. NQO-1 and HO-1 genes are downstream targets of Nrf-2, both of which are regulated by Nrf-2 expression, which is regarded as one of the most important intracellular antioxidant mechanisms [37,38]. Recent studies have reported that HO-1 and its metabolites have multiple biological functions, including anti-inflammatory, antioxidative, and antiapoptotic activities [39]. HO-1 is among the target genes of Nrf-2, which plays a vital role in inhibiting NF-κB induced by Nrf-2 [38]. Studies have shown that hydrogen molecules activate Nrf-2 in lung tissue, thereby promoting the expression of HO-1 and NQO-1 [40]. In our study, HRW treatment significantly elevated the expression of Nrf-2 and its downstream phase II metabolic enzymes, HO-1 and NQO-1, compared with the CON group. When the body’s antioxidant capacity cannot neutralize the toxic effects of LPS, the body’s antioxidant defense system collapses, and the regular expression of antioxidant proteins and genes cannot continue. As a result, intestinal cells are damaged and become necrotic, resulting in the release of TNF-α, heat shock proteins, ROS, and MPO, aggravating the inflammatory response and impairing endogenous antioxidant capacity [32,33,34]. NF-κB can also regulate Nrf-2-mediated ARE expression. A study by Yu et al. showed that p65 contributed to the abundance of nuclear Keap1 levels. In cells overexpressing p65, Keap1 reduced Nrf2-ARE signaling by translocating to the nucleus [41]. Therefore, compared with the control group, the reduction in the level of HO-1 and NQO-1 proteins might indicate the breakdown of the defense system. In the HRW + LPS group, HRW pretreatment upregulated the expression of HO-1 and NQO-1 proteins, suggesting that the body’s antioxidant capacity was enhanced. HRW might act on the Keap1/Nrf-2 pathway, inhibiting the binding of Keap1 and Nrf-2, promoting the expression of Nrf-2 in the nucleus, increasing the expression of HO-1 and NQO-1, and enhancing the body’s antioxidant capacity. In addition, SOD is the primary component of the defense mechanism against oxidative stress, participating in antioxidant defense by scavenging ROS [42], thereby preventing cellular damage. MDA content is an important parameter that can reflect the potential antioxidant capacity of an organism, also indirectly reflecting the degree of tissue peroxidative damage [43]. The effect of LPS on lipid peroxidation can be assessed by measuring the content of MDA in tissues. The results of the present study show that SOD levels were decreased in the LPS group, whereas MDA content was significantly increased. These results indicate that LPS inhibited the activity of SOD and induced lipid peroxidation, whereas HRW pretreatment enhanced SOD activity and downregulated MDA.

Oxidative stress and inflammation are interrelated in various pathological events. Previous studies have shown that the production of inflammatory cytokines is associated with the activation of NF-κB [44,45,46], which regulates the synthesis of TNF-α, IL-6, COX-2, etc. [47]. LPS is a commonly used proinflammatory factor in chronic inflammation models. After entering the body, LPS binds to the TLR4/MD-2 receptor and transduces the signal into the cell, activating the MyD88-dependent pathway and the NF-κB signaling pathway [48]. Yang et al. used LPS to stimulate endothelial cells and found that it increased the expression of TLR4 and activated the NF-κB signaling pathway [49], similar to our results. TLR4 and MyD88 protein expressions in rat ileum were significantly increased after LPS injection, indicating that TLR4 receptors and MyD88-dependent pathways were activated under the influence of LPS stimulation. In addition, LPS induction significantly upregulated the concentration and mRNA expression of IL-1β, IL-6, TNF-α, and MCP-1 in rat serum and ileal tissue, showing that LPS administration promotes the activation of the NF-κB signaling pathway and the expression of inflammatory factors. Additionally, COX-2, as a bifunctional enzyme, exhibits cyclo-oxygenase and peroxidase activity. COX-2 is expressed at low levels in most normal tissues [50]. When cells are stimulated by inflammation, the level of COX-2 increases in inflammatory cells, resulting in PGD2, PGE2, PGF2α, and PGI2 [51]. The overexpression of COX-2 has been described in the rat peritonitis model [52]. After LPS injection, COX-2 levels were significantly increased in response to inflammatory stimuli. Studies have also shown that the Nrf-2 signaling pathway has an anti-inflammatory effect, which is considered to be related to the inhibition of NF-κB. Recent studies have revealed the interaction between Nrf-2 and NF-κB. Furthermore, Nrf-2 has been found to act as an upstream signal of NF-κB during oxidative stress and inflammation [31,53,54,55]. In response to NF-κB, Nrf-2 acts as a protective mechanism against inflammation. Cuadrado et al. demonstrated that Nrf-2-mediated expression of HO-1 inhibited the proinflammatory activity of NF-κB [56]. Keap1–IKKβ interaction negatively regulates NF-κB by stabilizing IKBα [57]. Our results are consistent with those reported by Chen et al. [18]. In this study, we found that HRW inhibited LPS-induced NF-κB activation through the Nrf-2 signaling pathway and manifested as decreased IκBα phosphorylation levels and decreased NF-κB signaling pathway-related protein expression, thereby inhibiting the NF-κB signaling pathway. The activation of inflammatory cytokines reduced the expression of inflammatory factors, which is consistent with the results of proinflammatory cytokine concentrations and mRNA expression in the HRW + LPS group. COX-2 levels were also significantly reduced, indicating that HRW can also attenuate the inflammatory response by reducing COX-2.

LPS stimulation is known to activate the NF-κB signaling pathway, leading to an influx of chemotactic cytokines and peripheral circulating macrophages into the intestinal epithelial barrier [58]. In addition, studies have revealed that MCP-1 is related to the infiltration of monocyte and has the potential to activate monocytes/macrophages [59]. This process may produce chronic inflammation, damaging gut integrity and permeability. In the current study, we observed that LPS treatment resulted in a massive infiltration of macrophages and the accumulation of macrophages and neutrophils in the inflamed gut, which may create excess ROS and MPO by activating neutrophils to aggravate the inflammatory response [60,61]. Excessive ROS can destroy the structure of biological macromolecules, such as DNA, proteins, and lipids [62]. MPO content in the tissue also increased significantly. MPO, an enzyme involved in the oxygen-dependent microbial activity of phagocytes [63], has pro-oxidative and proinflammatory functions and is a biomarker of ROS-related injury [64]. In addition to oxidative effects, MPO affects various cell-signaling processes and inflammatory responses [65]. Increased levels of proinflammatory cytokines may be associated with MPO activity. In the HRW + LPS group, the lymphocyte infiltration level decreased, and ROS and MPO contents were also significantly reduced. These results show that HRW, as a potent antioxidant, could prevent cell damage, inhibit macrophage infiltration, and reduce tissue-destructive products produced by activated lymphoid, including proinflammatory cytokines, ROS, MPO, etc., reducing the inflammatory response of rat ileal tissue and enhancing the structure and function of the intestinal barrier to maintain the regular part of the intestine. In macrophages, the regulation of ROS and the associated signaling pathways could serve as potential targets for anti-inflammatory interventions. Therefore, HRW could reduce intracellular oxidative stress, control the inflammatory response, and prevent intestinal damage by activating the Nrf-2 signaling pathway.

Tight junctions consist of more than 30 structural or functional proteins [6]. They have complex protein structures comprised of transmembrane proteins, such as claudin-1 and occludin. They can interact with the actin cytoskeleton via plaque protein ZO-1 to regulate paracellular permeability. These three proteins also play a crucial role in maintaining the physiological functions of tight junctions [66,67]. Inflammation and pathogen invasion alter the expression of tight-junction proteins. Numerous studies have shown that upregulation of ZO-1 and occludin could increase intestinal permeability [68,69]. Therefore, the levels of occludin and ZO-1 are widely considered effective targets for treatment of intestinal diseases. To this end, we determined the expression of ZO-1, claudin-1, and occludin and measured the permeability of the intestine by OVA gavage in rats. Our results showed that LPS induction reduced ZO-1, claudin-1, and occludin but increased OVA levels in serum.

Additionally, severe disruption of ileal histology was observed following H&E staining, which showed that LPS-induced inflammation resulted in the impairment of small intestinal barrier integrity, consistent with the results of previous studies [70,71]. However, HRW treatment upregulated the expression of claudin, whereas the content of OVA in serum was significantly reduced, indicating that HRW exerted a protective effect on intestinal integrity. Previous studies revealed that HO-1 can upregulate tight-junction proteins and protect intestinal epithelial cells against oxidative damage [72]. Studies have also shown that the Nrf-2 pathway can activate HO-1 and increase the expression of ZO-1 and occludin, improving the anti-invasion ability of cells [73]. Therefore, we speculate that HRW can resist the invasion of inflammation and protect the intestine from damage.

## 5. Conclusions

In the present study, activation of the NF-κB signaling pathway by LPS resulted in intestinal and systemic inflammation accompanied by increasing intestinal oxidative stress and intestinal permeability. HRW increased the expression of tight-junction proteins, improving the antioxidant capacity of rats, reducing the content of ROS and MPO, alleviating the level of macrophage infiltration, activating the expression of the Nrf-2 signaling pathway, and regulating the NF-κB signaling pathway, thereby reducing LPS-induced intestinal chronic inflammation in rats, with positive antioxidant and anti-inflammatory effects.

## Figures and Tables

**Figure 1 vetsci-09-00621-f001:**
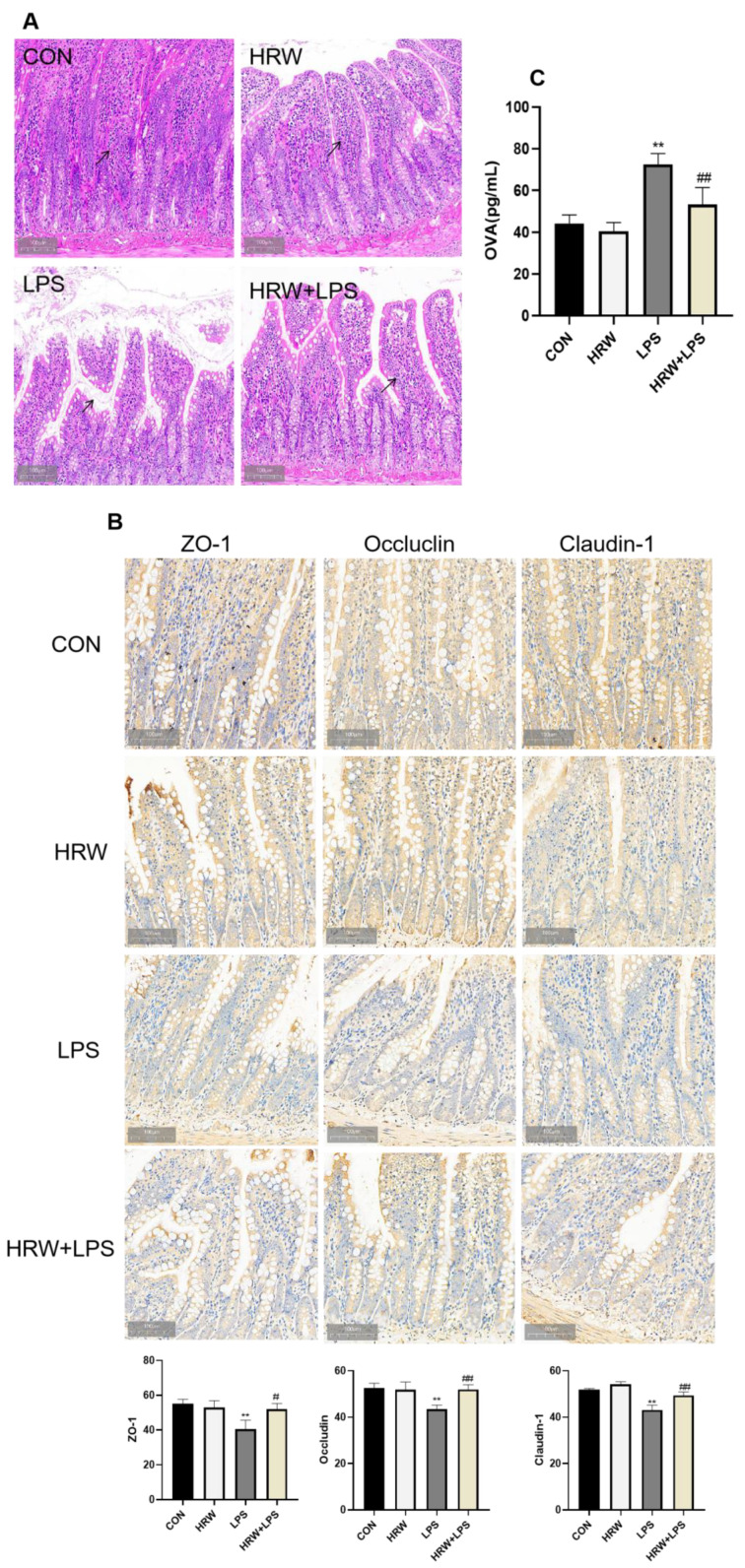
Effect of HRW on gut permeability. (**A**) Histopathological sections of rat ileum (150×). The black arrow indicates the villous state of the small intestine. (**B**) Immunohistochemical and quantitative analysis of ZO-1, occludin, and claudin-1 in ileal tissue. (**C**) OVA levels in the ileal tissue. # *p* < 0.05; **, ## *p* < 0.01.

**Figure 2 vetsci-09-00621-f002:**
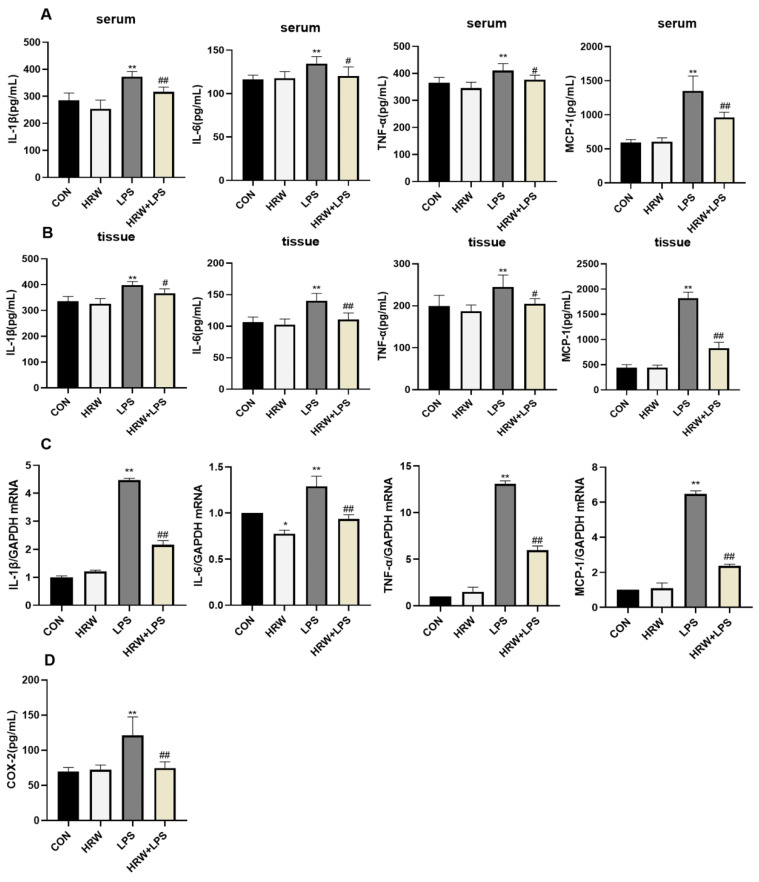
Effects of HRW on inflammatory cytokines in rats with chronic gut inflammation. (**A**) Serum levels of IL-1β, IL-6, TNF-α, and MCP-1. (**B**) Levels of IL-1β, IL-6, TNF-α, and MCP-1 in ileal tissue. (**C**) mRNA expression and activity of IL-1β, IL-6, TNF-α, and MCP-1. (**D**) Level of COX-2 in ileal tissue. *, # *p* < 0.05; **, ## *p* < 0.01.

**Figure 3 vetsci-09-00621-f003:**
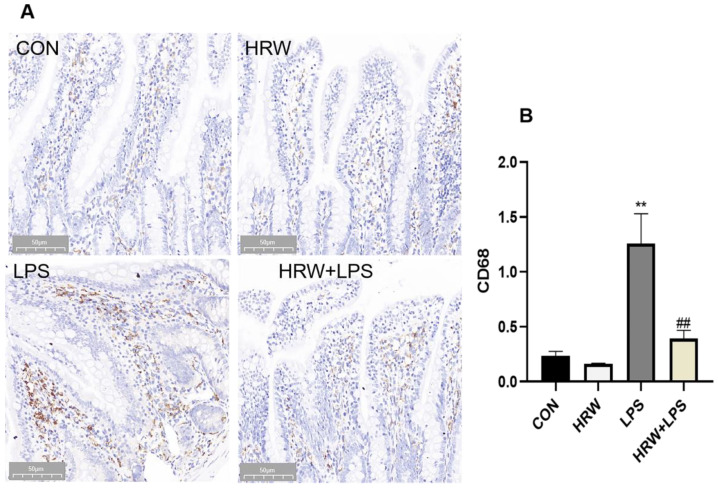
Effects of HRW on macrophage infiltration in rats with chronic gut inflammation. (**A**) Immunohistochemical analysis of macrophages in ileal tissue (200×). (B) Quantitative analysis of CD68 in ileum tissue. **, ## *p* < 0.01.

**Figure 4 vetsci-09-00621-f004:**
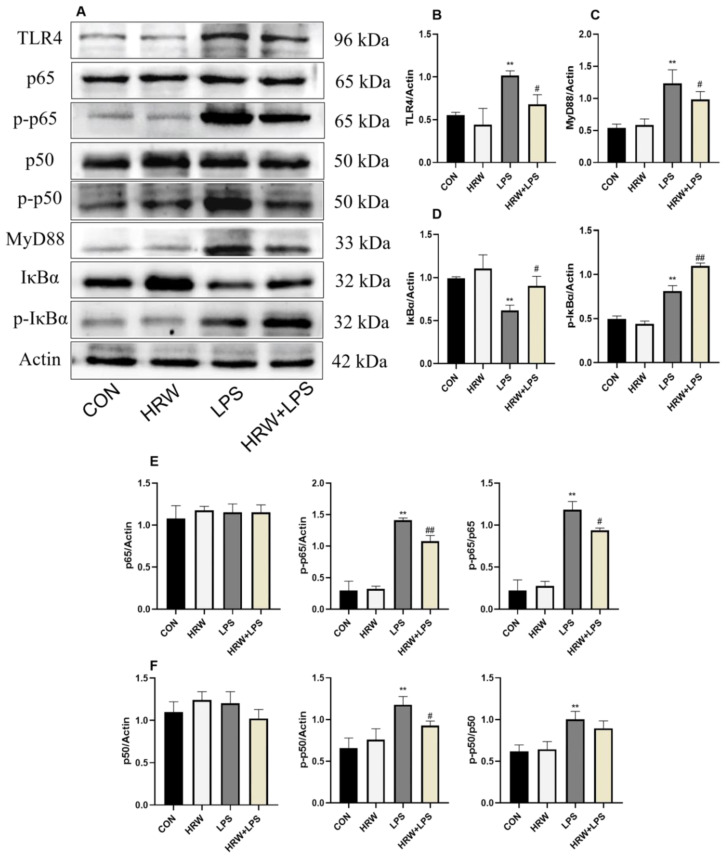
Effects of HRW on the NF-κB signaling pathway in rats with chronic gut inflammation. (**A**) Blot bands of protein expression related to the NF-κB signaling pathway. Protein expression of the NFκB signaling pathway was detected by Western blotting: (**B**) TLR4; (**C**) MyD88; (**D**) IκBα and p-IκBα; (**E**) p65 and p-p65 and their ratios; (**F**) p65 and p-p65 and their ratios. # *p* < 0.05; **, ## *p* < 0.01.

**Figure 5 vetsci-09-00621-f005:**
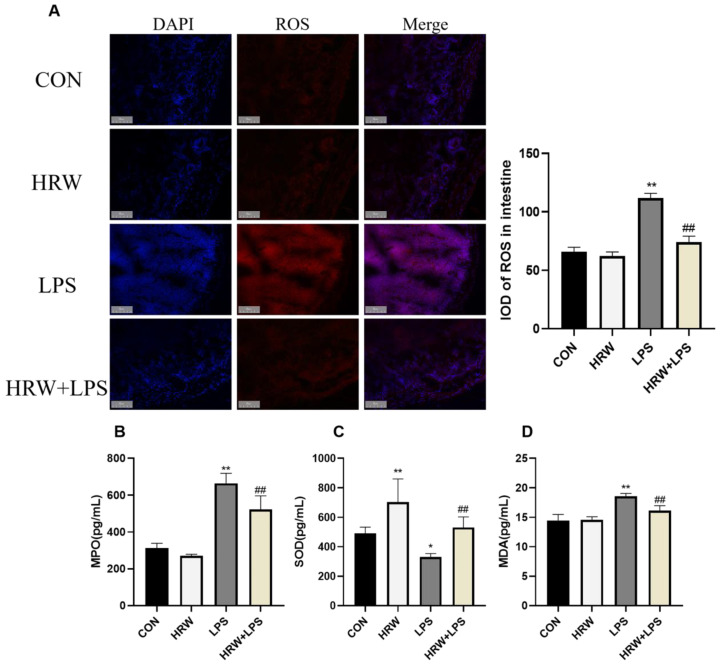
Effects of HRW on oxidative stress-related indicators in rats with chronic inflammation. (**A**) Immunofluorescence analysis and quantification of ROS in ileal tissue (200×). (**B**) Levels of MPO in ileal tissue. (**C**) Levels of SOD in ileal tissue. (**D**) Levels of MDA in ileal tissue. * *p* < 0.05; **, ## *p* < 0.01.

**Figure 6 vetsci-09-00621-f006:**
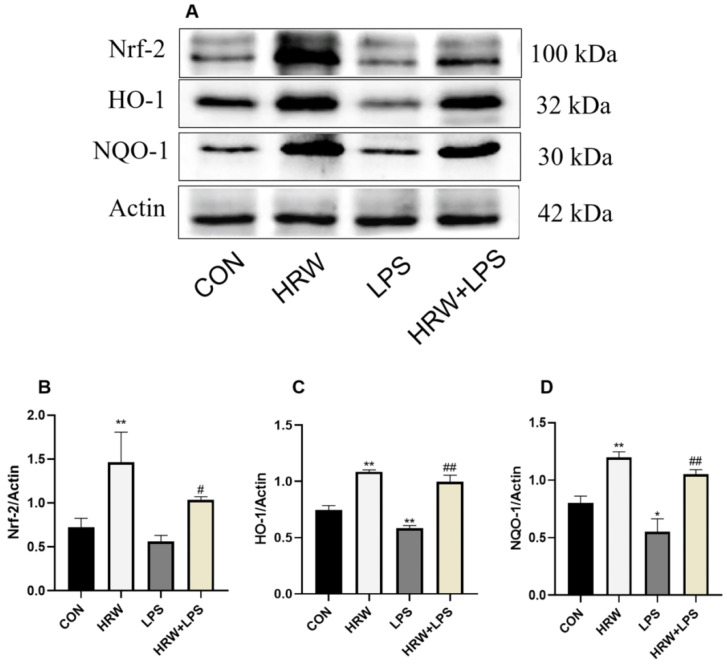
Effects of HRW on the Nrf-2 signaling pathway in rats with chronic gut inflammation. (**A**) Blot bands of protein expression related to the Nrf-2 signaling pathway. Protein levels of the Nrf-2 signaling pathway were detected by Western blotting: (**B**) Nrf-2; (**C**) HO-1; (**D**) NQO-1. * *p* < 0.05, ** *p* < 0.01 compared to the CON group; # *p* < 0.05, ## *p* < 0.01 compared to the LPS group. *, # *p* < 0.05; **, ## *p* < 0.01. Please view the original image in the Appendix A.

**Table 1 vetsci-09-00621-t001:** Primer sequences used in this study.

Gene	Gene Sequence Number	Primer Sequence (5′ to 3′)
GAPDH	NM_017008.4	AGGGCTGCCTTCTCTTGTG GGGTGGTCCAGGGTTTCTTAC
IL-1β	NM_031512.2	AATCTCACAGCAGCATCTCGACAAG TCCACGGGCAAGACATAGGTAGC
IL-6	NM_012589.2	ACTTCCAGCCAGTTGCCTTCTTG TGGTCTGTTGTGGGTGGTATCCTC
TNF-α	NM_012675.3	GCCTTGCCTTGCTGCTCTACC CTTCGTGGGGTTTGTGCTCTCC
MCP-1	NM_031530.1	CGCTTCTGGGCCTGTTGTTC CTCCAGCCGACTCATTGGGA

## Data Availability

I promise that all data are true and reliable. The datasets generated for this study are available upon request to the corresponding author.

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
