# Peer review of "Hydrogen-Rich Water Mitigates LPS-Induced Chronic Intestinal Inflammatory Response in Rats via Nrf-2 and NF-κB Signaling Pathways"

_vetsci, 2022, doi:10.3390/vetsci9110621_

Round 1
Reviewer 1 Report
1. Abstract: “This study investigated whether the oral administration of HRW is able to alleviate LPS-induced chronic intestinal” what is chronic intestinal?
2. “This study investigated whether the oral administration of HRW is able to alleviate LPS-induced chronic intestinal. In this study, an experimental model was established by feeding HRW in advance and injecting LPS (200 μg/ml) into the tail vein of rats.” “alleviate LPS-induced” and “model was established by feeding HRW in advance” are contradictory!
3. “related proteins and tight junction proteins. to study the antiinflammatory and antioxidant effects of HRW.” How to made this errors!
4. Introduction: It lacks a lot of supporting material to justify the current statement.
5. It is suggest that authors think about the implications of this experiment!
6. It there any Ethical Committee number?
7. Why to chose current drug concentration?
8. “2.3. Experimentaldesign” This manuscript is full of formatting and grammatical errors caused by carelessness, resulting in a lack of scientific rigor!
9. “* and # represent P < 0.05, indicating significant difference, ** and ## represent P < 0.01” It is funny! “P-value < 0.05 was considered statistically significant (* p < 0.05, ** p < 0.01)”
10. Numerous tests were performed to prove "Hydrogen-rich water alleviates LPS-induced chronic intestinal inflammatory response in rats via Nrf-2 and NF-κB signaling pathways". But the low level of writing obscures the scientific voice of this manuscript! It is strongly recommended to look for a native English speaker to modify the language. It is a major issue that must be resolved!
Author Response
请参阅附件。

Reviewer 2 Report
Dear authors,
Here my contribution to the final version of your article.
Abstract
This study investigated whether the oral administration of HRW is able to alleviate LPS-induced chronic intestinal ????.
Introduction
The major part for food digestion is locates in the small intestine which it is makes up a large portion of body’s external surface.
The injury of the intestinal barrier disrupts the metabolic balance of the gut and cause dysbiosis of the intestinal microbiota,
Recently, there is growing interest in Hydrogen (H2)
Materials and Methods
2.1. Animals
All rats were accommodated for 1 week before starting the experiments.
This study was approved by the Ethical Committee for Animal Experiments (Northeast Agricultural University, Harbin, China). Reference must be made to the serial number relating to the approval.
2.2. Drug concentration
Bibliographic references related to the referred methods must be cited.
2.3. Experimental design
The drinking time is limited at certain time points (8:00 to 10:00, 13:00 to 15:00, and 18:00 to 20:0).
D: group fed with hydrogen-rich water and injected with lipopolysaccharide through tail vein (HRW + LPS), fed HRW for 7 months, After 7 months, LPS was injected into the tail vein on the first day of every week, and the concentration was the same as that of the LPS group.
Group 4 underwent tail vein blood draw every weekend for 4 weeks.
Do you mean each animal from all groups?
In the groups to which LPS was administered, how was the induction of the intestinal inflammatory state proved?
2.4. Sample Collection
the rats were given ovalbumin (OVA). Please explain the purpose of that,...
After 4 weeks, the body weight of each group of rats was weighed and recorded. One hour before sacrifice, the rats were given ovalbumin (OVA) by gavage at a concentration of 250 mg·kg-1·bw. One hour after gavage, the rats were euthanized under general anesthesia. Blood was collected from the heart, centrifuged (10 min, 12 000 ×g, 4℃), and serum was stored at -80℃. The ileum was taken out, the intestinal contents were washed with PBS, and divided into two parts, one part was fixed with 10% formalin for histological, immunofluorescence, and immunohistochemical analysis. Other part ???
2.8. Histopathology
To evaluate the protective effect of hydrogen-rich water on intestinal injury, we per-formed pathological analysis of the ileum. After the rats were sacrificed, the intestinal tissues were preserved in 4% formaldehyde fixative solution,...
But previously you wrote: The ileum was taken out, the intestinal contents were washed with PBS, and divided into two parts, one part was fixed with 10% formalin for histological analysis,
It is important to mention the orientation chosen for the cuts.
All histological changes related to the loss of intestinal barrier integrity and to be studied in the histological images should be referred to in this section, eg presence and type of inflammatory infiltrate, .
Results
Figure 1 C seems has transversal instead longitudinal cut. All cuts should have same orientation to be compared.
In the HRW + LPS group, the structure of intestinal villi was significantly improved. When you compare with ???
Histological changes related to LPS administration should be noted on histological images.
3.2. Effects of HRW on inflammatory cytokines in the gut of rats with chronic inflammation
Compared with the CON group, the concentrations of IL-1β, IL-6, TNF-α, and MCP-1 in serum and tissues were sharply increased in the LPS group (Figure 2A, B, P < 0.01). However, the concentrations of IL-6, and TNF-α in the HRW + LPS group were decreased in serum compared with the LPS group (Figure 2A, P < 0.05). Also, the concentrations of IL-1β and MCP-1 were significantly ???? (Figure 2A, P < 0.01).
In intestinal tissue, compared with LPS group, IL-1β, and TNF-α content increased decreased (Figure 2B, P < 0.05) in HRW + LPS group, while the contents of IL-6 and MCP-1 decreased sharply (Figure 2B, P < 0.01).
Figure 2 should be previously announced in the text.
3.3. Effects of HRW on macrophage infiltration in the gut of rats with chronic inflammation
I can't find this parameter referred to in the material and methods section
3.4. Effects of HRW on NF-κB signaling pathway in rats with chronic inflammation
I can't find any reference to this parameter in the material and methods section, namely the proteins to be studied TLR4, MyD88,...IκBα, p-p50 and p-p65
Figure 5 - DAPI significance should be referred
3.6. Effects of HRW on Nrf-2 signaling pathway in rats with chronic inflammation
I can't find any reference to this parameter in the material and methods section, namely the proteins to be studied Nrf-2, HO-1 and NQO1
Discussion
LPS is a major component of the Gram-negative bacteria cell wall and is widely used for establishing the experimental models of inflammation in vitro and in vivo.
2.6. ELISA
Figure 6 legend - Levels of IL-1β, IL-6, TNF-α, MCP-1, Cyclooxygenase (COX-2), Myeloperoxidase (MPO), Ovalbumin (OVA), Malondialdehyde (MDA), but Malonic dialdehyde (MDA) content in discussion
Figure 6 legend - Levels of IL-1β, IL-6, TNF-α, MCP-1, Cyclooxygenase (COX-2), Myeloperoxidase (MPO),
In the current study, we observed that LPS treatment resulted in a massive infiltration of macrophages and the accumulation of macrophages and neutrophils in the inflamed gut, which may create excess reactive oxygen species (ROS) and myeloperoxdase enzyme (MPO) by activated neutrophils,which together aggravate the inflammatory response[54, 55].
MPO abbreviation is previously announced in figure 6 legend.
Our results showed that LPS induction produced the reduction of ZO-1, cladi-1 and occludin, but increased the OVA in serum.
Also, studies have shown that the Nrf-2 pathway can activate HO-1 and increase the expression of ZO-1 and occluding, which improves the anti-invasion ability of cells[69].
Round 2
Reviewer 1 Report
It can be accepted in the present version.
Author Response
Thank you very much for helping us to revise the manuscript.
Reviewer 2 Report
Title
Hydrogen-rich Water mitigates LPS-induced Chronic Intestinal Inflammatory Response in Rats via Nrf-2 and NF-κB Signaling Pathways
Abstract
Line 29 - NF-κB, but in Lines 23 and 24 - NFkB. Please keep the same notation.
As they chose to use abbreviations in the abstract and clarified the meaning of some, they should clarify the meaning of all of them as SOD.
Introduction
Line 45 - microbiota instead flora, please
Materials and Methods
2.11. Statistical analysis
Statistical significance values are due.
2.4. Sample Collection
the rats were given ovalbumin (OVA). Please explain the purpose of that,...in the text,....
Line 141 - Monocyte chemoattractant protein-1 (MCP-1)
Line 163 - Please explain the importance of determining these proteins. (TLR4, MyD88, IκBα, p-IκBα, p50, p-p50, p65, p-p65, Nrf-2, HO-1, and NQO1),
Results
Line 216 - There were also a large number of infiltrated inflammatory cells. Which cells ?
Line 217- "Compared with the LPS group, the structure of intestinal villi was significantly improved in the HRW + LPS group." Please describe,...
Statistical P in italics
Line 212 - Figure 1 A LPS seems has transversal instead longitudinal cut. All cuts should have same orientation to be compared. Please choose another photo with similar orientation to the others.
What do the arrows in Figure 1A mean?
Line 267 - P< 0.01.
Discussion
Line 321 - Use the expression "research works" instead of experiments, as the word experiments has a negative connotation.
Line 322 - Hydrogen molecules can be administered in multiple ways, including inhalation, oral and parenteral administration.
Line 340 - Musculoaponeurotic fibrosarcoma (Maf) proteins
AREs - What does that means?
Line 403 and 405 - IκBα (please keep the same abbreviations)
Line 412 - LPS stimulation is known to activate the NF-κB signaling pathway, leading to the influx of chemotactic cytokines and peripheral circulating macrophages into the intestinal epithelial barrier [59].
Line 424 - In addition to oxidative effects, lso
Line 451 - Reference 71 is related with gastric epithelial barrier and not intestine
Lines 452 to 453 - while the content of OVA in serum was significantly reduced.
but in Lines 466 to 468 - Our results showed that LPS induction reduced ZO-1, Claudin-1, and Occludin but increased the OVA in serum. ????
References
Please check all references. Journals names are not referred.
4. Luo P, Yang Z, Chen B, Zhong XJJoc, medicine m. The multifaceted role of CARD9 in inflammatory bowel disease. 494 2020;24(1):34-9.
Round 3
Reviewer 2 Report
Dear authors,
Congratulations.
Authors, reviewers and editors unite to communicate the best scientific works to society.